# Elucidation of Shearing Mechanism of Finish-type FB and Extrusion-type FB for Thin Foil of JIS SUS304 by Numerical and EBSD Analyses

**DOI:** 10.3390/ma12132143

**Published:** 2019-07-03

**Authors:** Yohei Suzuki, Tomomi Shiratori, Ming Yang, Masao Murakawa

**Affiliations:** 1Komatsuseiki kosakusho. Co., Ltd., 942-2 Shiga, Suwa, Nagano 392-0012, Japan; 2Tokyo Metropolitan University, 6-6, Asahigaoka, Hino, Tokyo 191-0065, Japan; 3Nippon Institute of Technology, 4-1 Gakuenndai, Miyashiro, Minamisaitama, Saitama 345-8501, Japan

**Keywords:** fine blanking, metallic microgear, finite element analysis, electron backscatter diffraction, critical fracture value

## Abstract

A numerical analysis using FE (finite element) analysis was performed to clarify the shearing mechanism in the process of extrusion-type fine blanking (FB) for a thin foil of JIS SUS304 in this study. Extrusion-type FB, in which a negative clearance between the punch and the die has been developed and investigated experimentally to improve the quality of the sheared surface in the blanking of thin foils. The resultant sheared surface for extrusion-type FB indicated an almost completely sheared surface, and the fracture portion on the sheared surface was much smaller than that in conventional FB, the so-called finish-type FB. The material flow and fracture criteria in extrusion-type FB were analyzed in comparison with those in finish-type FB. The differences in material flow and so-called critical fracture value were verified for the two processes. The principal stress near the shearing surface has mostly compressive components in extrusion-type FB due to its negative clearance, and the critical fracture value was also less than that in finish-type FB, in which the principal stress near the shearing surface has mostly tensile components. Furthermore, SEM observation with EBSD (electron back-scatter diffraction) analysis of the shearing surface was performed to verify the phenomena. Reductions in deformation-induced crystal orientation rotation and martensite transformation in extrusion-type FB were confirmed in comparison with those in finish-type FB from the analysis results.

## 1. Introduction

Recently, the notable miniaturization of electronic devices and medical instruments has been observed. Accordingly, the miniaturization of their components and their high-precision capability is also required [1]. The fabrication of microparts with complex profiles using stamping press machines is attractive owing to its high productivity. Blanking is one of the typical stamping processes used to rapidly cut parts with complex profiles from sheet metal, and fine blanking (FB) [2] can be used to make a part with a fine shearing surface. However, problems due to the miniaturization of parts arise in microblanking. As parts become smaller, the sheet becomes thinner. In general, the clearance between the punch and the die is expressed as the ratio to the thickness of the sheet, so the absolute value of the clearance also decreases as the sheet thickness decreases [3]. According to findings on the FB process at the macro scale, the clearance is assumed to less than 1% of the thickness *t* [4]. Since a sheet of 0.1 mm thickness or less is usually used in microblanking, the clearance becomes 1 μm or less and, as a consequence, the manufacturing accuracy of the die and the positioning accuracy of the punch and die are approaching their limits [5]. Furthermore, when the thickness of the work material decreases, fracture is likely to occur owing to the effects of the surface roughness of the material and the crystal grain size [6,7,8,9], which affects the product accuracy. To solve these problems, the authors previously developed a novel FB process and demonstrated its advantages by an experimental comparison with the conventional FB process using a general-purpose precision stamping press machine [10]. The conventional FB process is finish-type FB with an extremely narrow positive tool clearance and a counterpunch. The newly developed process is extrusion-type FB with a negative tool clearance and a counterpunch pressure pad. By comparing the results, the developed process was found to be superior from the most important viewpoint of suppressing the fracture of the surface. However, the material flow and fracture mechanism have not yet been clarified, and furthermore, the most important parameters in the process are unknown.

In this study, an FE analysis is performed to clarify the material flow and fracture mechanism in extrusion-type FB in comparison with finish-type FB, particularly from the viewpoint of suppressing fracture or cracking on the shearing surface during the FB process. Furthermore, the mechanism of fracture suppression was verified by SEM observation with EBSD (electron back-scatter diffraction) analysis, which enables the observation of the material deformation state using the information obtained by the orientation analysis of the crystallinity of the shearing surface of samples.

## 2. FEM Simulation Model and Conditions 

Figure 1 shows SEM images of microgears fabricated by finish-type FB and extrusion-type FB [10]. As described previously, the shearing surface of the microgear processed by extrusion-type FB is smoothly sheared without fracture in comparison with that processed by finish-type FB. 

To elucidate the mechanism of fracture suppression, we developed an analysis model for both FB methods for use in FE simulation, and the model and its conditions are shown in Figure 2 and Table 1, respectively. The microgear was assumed to be axisymmetric about the gear centerline, and a two-dimensional model was applied in this simulation to reduce the calculation time. The commercially available FEM (Finite element method) code DEFORM2D (Ver.11.1) was utilized for the simulation. The punch force *Fp* was processed as a constant displacement type, i.e., *Fp* was calculated as the reaction force *Fm* of the workpiece. The workpiece used was a JIS SUS304 sheet with a thickness t of 0.178 mm, and a microgear workpiece having an outer diameter of *φDw* of 3.5 mm was used to correspond to the experiment at conditions of the press in the previous report [10]. In our present FE analysis, the outer peripheral edge of the workpiece was assumed to be constrained, as shown in Figure 2, since a coil material having sufficient width relative to the size of the microgear was used in the experiment, and there was no material flow in the width direction of the material. A die diameter *φD_d1_* of 1.752 mm (tool clearance *Cl* of 2 μm) was used for finish-type FB, and a die diameter *φD_d2_* of 1.732 mm (clearance *Cl* of −8 μm) was used for extrusion-type FB. The punch diameter *φDp* of 1.748 mm and the counterpunch diameter *φDc* of 1.720 mm were common to both processes. The punch, die and blank holder/stripper were assumed to be rigid bodies, and the workpiece was assumed to be elastoplastic. Approximately 15,000 four-node rectangular elements were generated on the workpiece. Since the deformation is concentrated in an extremely narrow range in the deformation area around the cutting edge of the tool, there is significant distortion in the elements [11]. The simulation result in a previous FB study, which revealed the relationship between the mesh size and the clearance *Cl*, was referred to for the determination of the mesh size [12]. Specifically, the mesh size in the analysis was set to about 1 μm, which is smaller than the clearance *Cl*. In such a case where the deformation of the elements was apparently large, a remeshing function was adopted to rebuild the meshes. 

The material constants were derived from the data available from an actual experimental tensile test giving the flow stress-plastic strain curve shown in Figure 3. Furthermore, to verify and observe the fracture process in both FB methods, we used the damage value C (critical fracture value [13]) obtained from the ductile fracture criterion [14] of Cockcroft and Latham defined as the value of the integral corresponding to the evolution of the maximum principal stress as follows: (1)$$C\;=\int_{0}^{\bar{\varepsilon }}\frac{{\bar{\sigma }}_{max}}{\bar{\sigma }}d\bar{\varepsilon }$$
where *C*: critical fracture value, ${\bar{\sigma }}_{max}$: maximum principal stress, $\bar{\sigma }$: effective stress, and $\bar{\varepsilon }$: effective strain.

The friction coefficient was assumed to be 0.08, referring to the value recommended by DEFORM2D for cemented carbide dies. Although there was a concern that the friction may change significantly during the blanking process, as a result of examining them in the report of Sasada et al. [15]. It was reported that the FEM analysis agrees well with the blanking result even if the friction coefficient was constant. Accordingly, the coefficient of friction was set constant in this study. The temperature was set to be constant in this analysis. Although temperature changes occur in the tool and material during blanking, the ratio of heat flow to the material could be considered negligibly small owing to the small volume ratio of the material to the die at the microscale. 

## 3. FEM Simulation Results and Discussion

Since fracture occurred at a punch stroke of about 80% of the sheet thickness t according to our previous experimental results [10] for finish-type FB, the damage values (critical fracture values) at a punch stroke of 80% were investigated for both processes. Figure 4 shows a comparison of the damage value between a) finish-type FB and b) extrusion-type FB in the case of the punch penetration. In the case of finish-type FB, the result shows that damage accumulated in the region connecting the punch shoulder and die. In the case of extrusion-type FB, the damage value was also high in the punch shoulder and punch side, but it was lower in other regions including the shearing zone, which means that some allowance remains to prevent fracture. This was in agreement with the experimental results obtained with an actual stamping press machine in which extrusion type FB did not cause any breakage or separation of the workpiece, even at 80% punch penetration through the sheet thickness *t* [10].

The von Mises stress and maximum principal stress distributions for both FB methods are shown in Figure 5 and Figure 6, respectively, for the same condition of punch penetration of 80% as in Figure 4. It was found that, although the von Mises stresses for both FB methods showed similar distributions, the maximum principal stress distributions were significantly different. It can be observed that, in the case of finish-type FB, in the area between the scrap and the product (being blanked out), a tensile stress distribution appears to prevail, while in the case of extrusion type FB, a compressive stress distribution prevails. These results are consistent with the results for the damage value shown in Figure 4, in which higher maximum principal stresses corresponded to higher damage values, consistent with previous knowledge that the compression stress distribution can reduce the damage value and suppress fracture in blanking processes even if the von Mises stress is similar [16].

From the various reports thus far, it is known that the stress distribution within a material is related to the material flow in the processing. For example, Huang et al. have shown the effect of the tool clearance (either positive or negative) on the die roll during blanking [17], and Thipprakmas et al. have clarified the mechanism of fracture during FB, in which an extremely small but positive tool clearance was used [18]. However, they have not yet clarified the fracture mechanism, in which a positive clearance and a negative clearance might cause different fracture behaviors. The material flows in both FB processes were also compared at the same punch penetration of 80% sheet thickness and Figure 7 shows the resultant material flows in the shearing zone. In the case of finish-type FB, the workpiece located immediately under the punch flows almost perpendicularly to that in the die interior portion, inducing material flow on the scrap side that was parallel to the interior material flow. Accordingly, we concluded that this material flow behavior generates a tensile stress field between the scrap side of the workpiece (which is pressed down by the blank holder) and the blanked-out workpiece, resulting in a consequently high cumulative damage value, ultimately leading to the breakage of the workpiece. In the case of extrusion type FB, a change in the direction of the workpiece flow is observed near the die corner B. In particular, immediately adjacent to point B, owing to the material flow towards the die portion B, the workpiece behaves as if it were self-crushed. As a result, we consider that a compressive stress field is generated where fracture is suppressed.

## 4. Discussion Based on EBSD Analysis

The shearing surfaces were observed by SEM with EBSD (electron back-scatter diffraction) to clarify the shearing mechanism for both FB methods. EBSD analysis is an effective methodology for evaluating the deformation state of grains in a workpiece. In particular, the KAM (kernel average misorientation) is a value that indirectly represents the degree of plastic deformation via the amount of crystal orientation and can correspond to the equivalent plastic strain in plastic working [19]. Furthermore, it has been used to clarify the fracture generation mechanism during microshear processing [20]. Since it is known that JIS SUS304 transforms from a ductile austenite phase to a brittle martensitic phase depending on the processing strain, information on the phase transformation will be also useful for understanding the fracture mechanism.

For sample preparation for EBSD analysis, first of all, the cross section of workpieces of the microgear half-blanked by the punch that penetrated about 60% of the sheet thickness for both FB methods was fabricated. Next, it was cut to the area shown in Figure 8, and the surface was mirror-finished by mechanical polishing and ion milling. EBSD analysis was performed at an acceleration voltage of 20 kV and a measurement interval of 0.15 μm. Figure 9 shows the analysis results in the form of KAM maps for the finish and extrusion-type FB methods. Figure 10 shows the analysis results in the form of phase maps. It was found that the phase transformation from the austenite phase (red area) to the martensitic phase (green area) occurs in the areas with high strain in the KAM maps. It was found that areas with a high strain of five degrees red areas and different crystal orientations are distributed in the area connecting the punch edge and the die corner. Moreover, the phase transformations occurred in areas with high strain in both methods consistent with the results of FE analysis. However, the areas with high strain and phase transformation were concentrated in the shearing zone in the case of finish-type FB, while they shifted to the scrap side (left side) for extrusion-type FB. In these areas, compression stress is dominant and does not contribute to fracture, as described in the last section. By assuming that cracks occur along the vertical shearing line (white lines in Figure 9 and Figure 10), a difference in strain concentration on the white line (length of about 80 μm) can be seen for the two methods. Figure 11 shows the KAM values along the line from point a to point b in Figure 9 for the two methods. In finish-type FB, the KAM values are mostly five degrees from point a to point b. On the other hand, in extrusion-type FB, the KAM values along Y axis are approximately two degrees around point a and increase toward point b. These findings indicate that strain has already accumulated in the area where shearing is going to proceed in finish-type FB, while there is less accumulation of strain in the area in extrusion-type FB. As a result, extrusion-type FB can prevent fracture from the viewpoint of the strain accumulation region and crystal grain phase field. 

From the discussion in Section 3 and Section 4, we can conclude that the extrusion-type FB process can prevent fracture and improve the blanking quality by changing the material flow during the process. It is a high-potential process for blanking at the microscale. However, since the material is compressed in the negative clearance zone in addition to the shearing, the load on the punch in extrusion-type FB is higher than that in conventional FB. It is necessary to clarify design guidelines for the process parameters, such as the optimal negative clearance and the radius of the die corner, in further work.

## 5. Conclusions

An FE analysis was performed to clarify the material flow and fracture mechanism in extrusion-type FB, in which a negative clearance was applied, in comparison with that in finish-type FB, particularly from the viewpoint of suppressing the fracture and cracking on the shearing surface during the process. The critical damage value, an index showing the possibility of fracture in the workpiece during FB, was introduced as a critical fracture value. Furthermore, the SEM observation of the cross-section of blanked workpieces with EBSD analysis was performed to confirm the results of the analysis. The conclusions of this study are as follows:

(1) The perimeter of the workpiece under a punch flows almost perpendicularly towards the die interior in finish-type FB. This workpiece flow behavior generates a tensile stress field between the scrap-side workpiece (which is pressed down by the blank holder) and the blanked workpiece, with a consequent high damage accumulation value, ultimately leading to breakage of the workpiece.

(2) In contrast, in the case of extrusion-type FB, since the perimeter of the workpiece flow under the punch flows towards the end of the die radius, high compressive stress occurred in this region, which led to different material flow and reduced cumulative damage. 

(3) The critical damage value for extrusion-type FB was less than that for finish-type FB owing to the difference in stress state near the shearing zone: compressive stress is dominant for extrusion-type FB, while tensile stress is dominant for finish-type FB.

(4) The difference in strain distribution between finish-type FB and extrusion-type FB was analyzed by EBSD. The relationship between the strain distribution and the occurrence of fracture was clarified.

## Figures and Tables

**Figure 1 materials-12-02143-f001:**
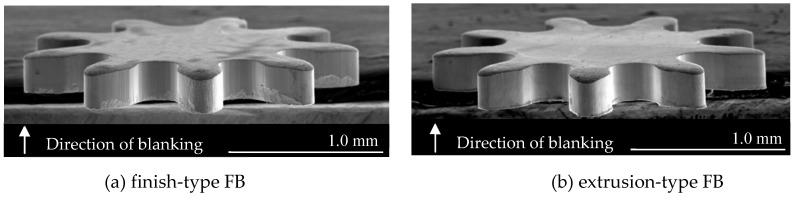
SEM images of the microgear [10]: (**a**) finish-type FB; (**b**) extrusion-type FB.

**Figure 2 materials-12-02143-f002:**
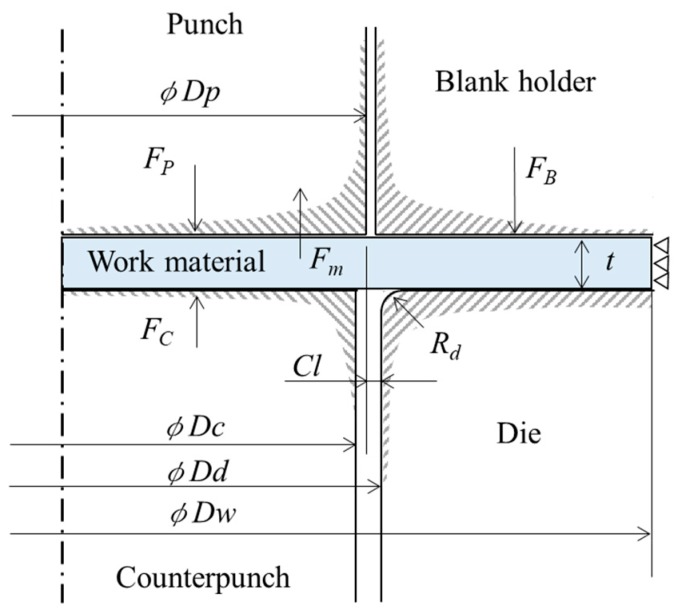
FEM simulation model.

**Figure 3 materials-12-02143-f003:**
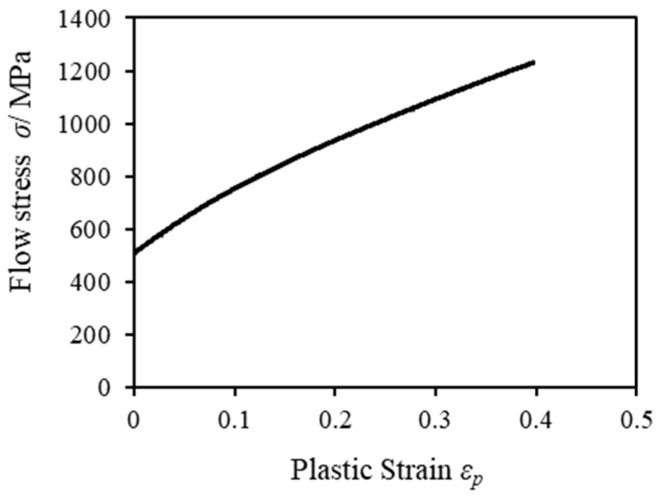
Flow stress-plastic strain curve.

**Figure 4 materials-12-02143-f004:**
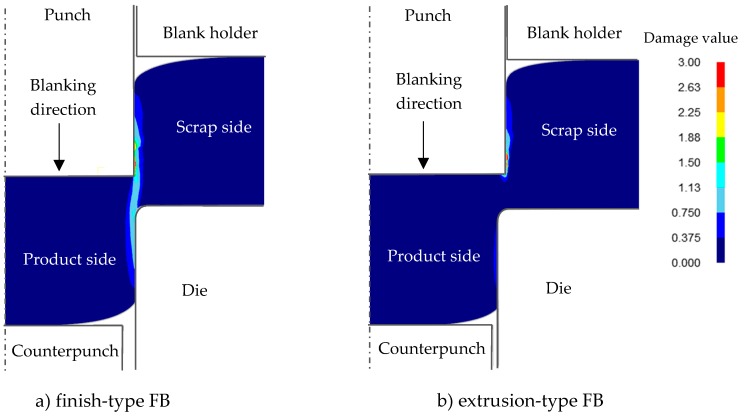
Comparison of damage value defined as the integration value corresponding to the evolution of the maximum principal stress ${\bar{\sigma }}_{max}$ for 80%*t* punch penetration between finish-type fine blanking (FB) and extrusion-type FB: (**a**) finish-type FB; (**b**) extrusion-type FB.

**Figure 5 materials-12-02143-f005:**
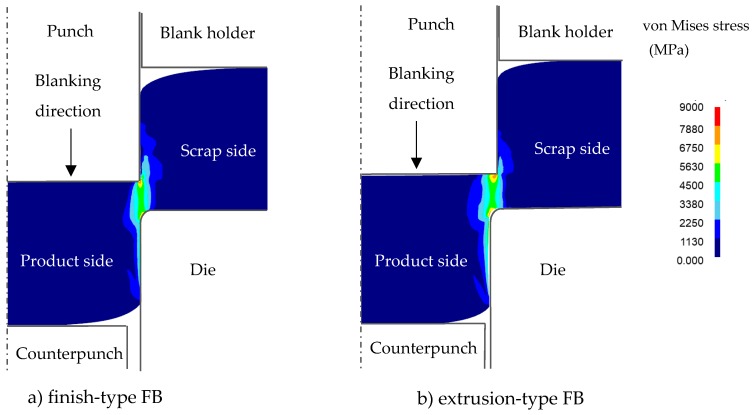
Comparison of von Mises stress for 80%*t* punch penetration between the finish-type FB and extrusion-type FB in Figure 4: (**a**) finish-type FB; (**b**) extrusion-type FB.

**Figure 6 materials-12-02143-f006:**
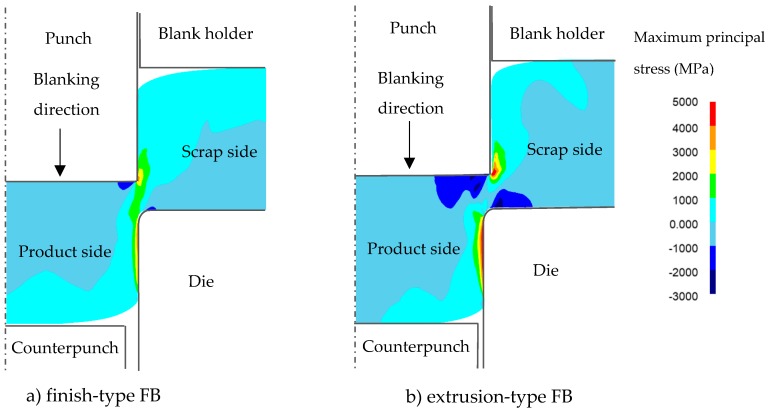
Comparison of maximum principal stress for 80%*t* punch penetration between the finish-type FB and extrusion-type FB in Figure 4: (**a**) finish-type FB; (**b**) extrusion-type FB.

**Figure 7 materials-12-02143-f007:**
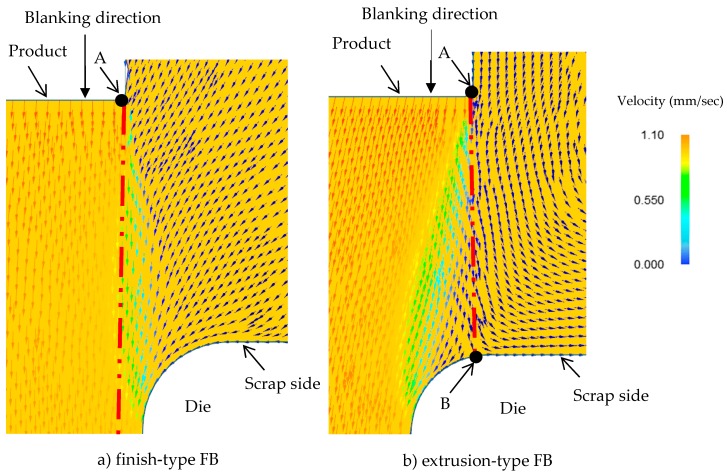
Comparison of material flow for 80%*t* punch penetration between finish-type FB and extrusion-type FB, in an enlarged view: (**a**) finish-type FB; (**b**) extrusion-type FB.

**Figure 8 materials-12-02143-f008:**
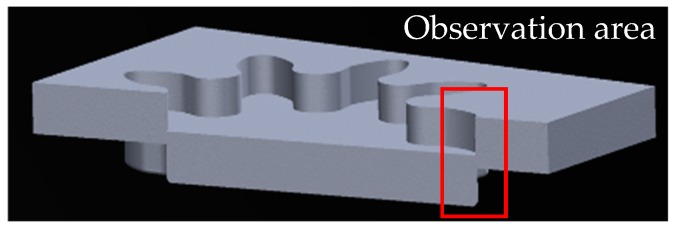
Observation area of microgear for electron back-scatter diffraction (EBSD) analysis.

**Figure 9 materials-12-02143-f009:**
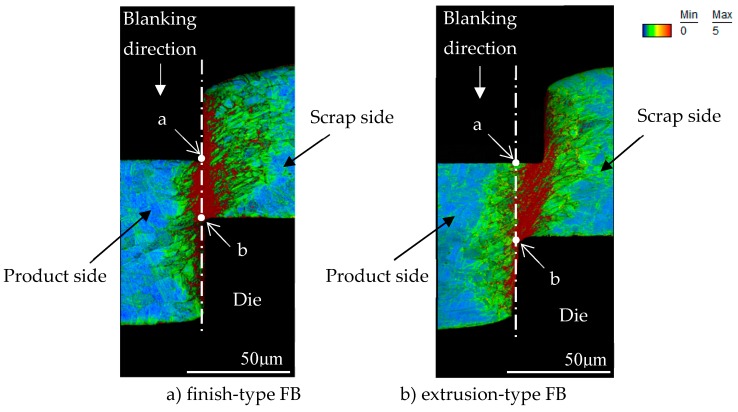
Comparison of kernel average misorientation (KAM) map for 60%*t* punch penetration between finish-type FB and extrusion-type FB: (**a**) finish-type FB; (**b**) extrusion-type FB.

**Figure 10 materials-12-02143-f010:**
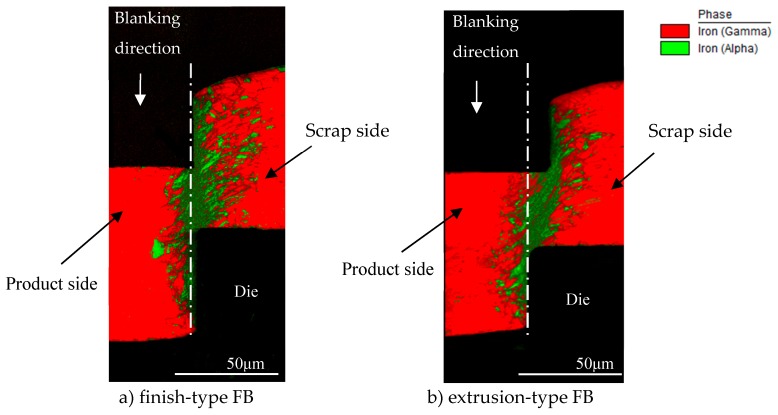
Comparison of phase map for 60%*t* punch penetration between finish-type FB and extrusion-type FB: (**a**) finish-type FB; (**b**) extrusion-type FB.

**Figure 11 materials-12-02143-f011:**
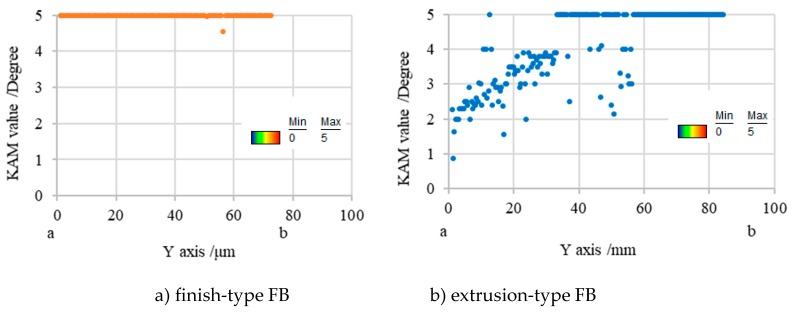
Comparison of KAM values between a and b of Figure 7: (**a**) finish-type FB; (**b**) extrusion-type FB.

**Table 1 materials-12-02143-t001:** FEM simulation conditions.

Simulation Model Type	Axisymmetric Model
Object type	Workpiece: elasto-plastic (*φD_w_*:3.5 mm)
	Punch /Die: rigid (*φD_p_*:1.748 mm, *φD_d1_*:1.752 mm, *φD_d2_*:1.732 mm)
	Blank holder /Stripper: rigid
	Counterpunch: rigid (*φD_c_*:1.720 mm)
Clearance (*Cl)*	Finish-type FB: 2 μm
	Extrusion-type FB: −8 μm
Blank holder force (*F_B_*)	1000 N (50% of maximum blanking force)
Counterpunch force (*F_C_*)	400 N (20% of maximum blanking force)
Blanking force (*F_P_*)	Non-constant value
Radii of tool cutting edges	*R_p_* = 0.00 mm, *R_d_* = 0.01 mm
Work material (Workpiece)	SUS304 *t* = 0.178 mm Young’s modulus: 193 GPa Poisson’s ratio: 0.3
Fracture criterion equation	Cockcroft and Latham
Friction coefficient (*μ*)	0.08

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
