# Peer review of "Elucidation of Shearing Mechanism of Finish-type FB and Extrusion-type FB for Thin Foil of JIS SUS304 by Numerical and EBSD Analyses"

_materials, 2019, doi:10.3390/ma12132143_

Reviewer 1 Report

The paper brings significant information to manufacturing processes literature, is well organised and the results are well explained.

Author Response

We wish to express our appreciation to your comments on our paper.

Reviewer 2 Report

Since the paper discusses an FE approach to understand fracture mechanisms, please provide a more comprehensive review of the literature with respect to FB in the introduction.

While the experimental details for manufacturing the microgear used in the EBSD analysis is referenced in the paper, it would be very beneficial to recount that briefly in this paper as well.

Author Response

Point 1 Since the paper discusses a FE approach to understand fracture mechanisms, please provide a more comprehensive review of the literature with respect to FB in the introduction.

Response 1: We added comprehensive review of the literature with respect to FB in the introduction as [2]. Accordingly, we changed the numbers of other references.

Point 2While the experimental details for manufacturing the microgear used in the EBSD analysis is referenced in the paper, it would be very beneficial to recount that briefly in this paper as well.

Response 2: We explained the analysis method of EBSD in P7, after the 259th line.

Reviewer 3 Report

This paper explored the shearing mechanism of two interesting manufacturing processes. The fundamental mechanics was clearly presented using both numerical simulation and experimental techniques, which is a good work. There are several minor questions from the reviewer:

Typo in line 47: theier --> their.

The extrusion-type FB with a negative clearance of -8 μm can provide superior part quality than the typical finish-type FB. Can the author provide any information of tool wearing for these two processes? When the other factor such as tool wearing is considered, is this clearance the optimum? 

The friction may vary significantly during the blanking process. In the simulation, a constant coefficient of friction was used. Can the author provide any parametric study to see the effect of COF?

Line 171: Is the equivalent stress von Mises type? If so, please clarify.

In Figure 6, the Max. principal stress was used to demonstrate that the compressive stress prevails in the extrusion-type FB. Would it make more sense to show the Min. principal instead?

Author Response

Point 1:Typo in line 47: theier --> their.

Response 1: Thank you for pointing it out. We amended it.

Point 2:The extrusion-type FB with a negative clearance of -8 μm can provide superior part quality than the typical finish-type FB. Can the author provide any information of tool wearing for these two processes? When the other factor such as tool wearing is considered, is this clearance the optimum?

Response 2: We certainly understand your comment. However, since the purpose of this research is to clarify the shearing/fracture mechanism, although we have relevant data such as tool wear and tool load for the -8μm clearance, the optimum clearance in question will be a future work.

Point 3:The friction may vary significantly during the blanking process. In the simulation, a constant coefficient of friction was used. Can the author provide any parametric study to see the effect of COF?

Response 3: We added a reference [15] that investigated the coefficient of friction between the tool and material in blanking process, and performed FEM analysis using the results. The reference showed that the FEM analysis agrees well with the press experiment results even if the friction coefficient is set constant. Therefore, the coefficient of friction was set constant in our study. We explained it in P3, after the 132th line.

Point 4:Line 171: Is the equivalent stress von Mises type? If so, please clarify.

Response 4: Yes, it is. We clarified as follows.  von Mises stress.

Point 5:In Figure 6, the Max. principal stress was used to demonstrate that the compressive stress prevails in the extrusion-type FB. Would it make more sense to show the Min. principal instead?

Response 5: The damage value shown in Figure 4 was determined from the maximum principal stress. Therefore, we used the maximum principal stress from the relationship.